# Insights in Post-Translational Modifications: Ubiquitin and SUMO

**DOI:** 10.3390/ijms23063281

**Published:** 2022-03-18

**Authors:** Daniel Salas-Lloret, Román González-Prieto

**Affiliations:** 1Cell and Chemical Biology, Leiden University Medical Centre, Einthovenweg 20, 2333ZC Leiden, The Netherlands; D.salas@lumc.nl; 2Genome Proteomics Group, Department of Genome Biology, Andalusian Centre for Regenerative Medicine and Molecular Biology (CABIMER), Américo Vespucio 24, 41092 Seville, Spain

**Keywords:** ubiquitin, SUMO, E3 enzymes, proteomics

## Abstract

Both ubiquitination and SUMOylation are dynamic post-translational modifications that regulate thousands of target proteins to control virtually every cellular process. Unfortunately, the detailed mechanisms of how all these cellular processes are regulated by both modifications remain unclear. Target proteins can be modified by one or several moieties, giving rise to polymers of different morphology. The conjugation cascades of both modifications comprise a few activating and conjugating enzymes but close to thousands of ligating enzymes (E3s) in the case of ubiquitination. As a result, these E3s give substrate specificity and can form polymers on a target protein. Polymers can be quickly modified forming branches or cleaving chains leading the target protein to its cellular fate. The recent development of mass spectrometry(MS) -based approaches has increased the understanding of ubiquitination and SUMOylation by finding essential modified targets in particular signaling pathways. Here, we perform a concise overview comprising from the basic mechanisms of both ubiquitination and SUMOylation to recent MS-based approaches aimed to find specific targets for particular E3 enzymes.

## 1. Introduction

The environment is constantly changing. Consequently, cells need to adapt and give a quick and appropriate response against different stimuli or stresses. Proteins control the vast majority of cellular processes, and their function is regulated by a variety of post-translational modifications (PTMs). Commonly, PTMs are covalent enzymatic modifications at the amino acid chain or termini of proteins that may happen after translation and can affect the function of a target protein, including the activity, folding, localization, interaction partners, and protein homeostasis. PTMs can be completed in seconds [1] and are generally reversible by the action of specific deconjugating enzymes, which enable the cells to provide a rapid and precise response to environmental changes. More than 200 different PTMs have been described, which can either consist of the addition of a small chemical group (e.g., acetylation, methylation, phosphorylation, etc.), the modification by complex molecules (i.e., glycosylation, AMPylation, ADPribosylation, prenylation, etc.) or the addition of long polypeptide chains such as ubiquitin or ubiquitin-like modifiers (UBLs). Additionally, irreversible PTMs exist (i.e., proteolysis, deamidation, etc.) and recent research has reported the role of deamidation in the regulation of ubiquitin E3 enzymes [2] (Figure 1).

In this review, we provide a general overview of the current tendencies in the ubiquitination and SUMOylation fields, discussing the basic principles of these modifications and their importance in complex signaling pathways. We compare the enzymes involved in the catalytic cycle of both PTMs and analyze different approaches to identify target proteins of these modifications. Finally, we discuss the future directions in the field with the use of new inhibitors, genome-wide screenings, and MS-based proteomics approaches.

## 2. Ubiquitination

Ubiquitination, namely, the covalent attachment of ubiquitin (Ub) to acceptor residues in target proteins is, after phosphorylation, the second PTM in abundance [3] and regulates virtually all events in cells. 

Ubiquitin received its name from being ubiquitous in all cell types among eukaryotes. It is a highly conserved 76 amino acids protein which differs by only 2 amino acids from yeast to humans and by 3 amino acids from plants to humans. However, Ub has not been identified in *procaryotes*. Although, its origins may reside in bacteria protein molybdopterin converting factor subunit 1(MoaD) and thiamine biosynthesis protein S (ThiS). These proteins share the same structure with Ub but are mainly involved in biosynthetic pathways [4,5]. Years later, other bacterial proteins such as the small prokaryotic ubiquitin-like protein (Pup) from *Mycobacterium tuberculosis* have also been described. Similar to Ub, Pup becomes conjugated to 26S proteasome targets, sharing comparable function to ubiquitin in eukaryotes [6]. The prokaryotic ancestor may not only create Ub, but also ubiquitin-like proteins including SUMO, NEDD8, and ISG15. In total, 20 human UBLs have been reported to be conjugated to other molecules. They share a common overall fold and a 3-step conjugation cascade consisting of their own E1, E2, and E3 enzymes. Despite their similarity, they impart distinct functions to their substrate proteins [7] and participate together coordinating different cellular functions [8,9,10,11]. 

Most eukaryotic genomes contain multiple Ub genes. In humans, Ub is encoded by four genes: UBB, UBC, UBA52, and RPS27A. UBA52 and RPS27A encode single copies of Ub, while UBB and UBC encode polyubiquitin chains that are cleaved by the Ubiquitin Specific Protease USP5 to produce monomeric active Ub molecules. These monomeric active Ub molecules fold into a compact and globular structure with a terminal diglycine (diGly) sequence essential for its conjugation [12].

During the ubiquitination cascade, the activating enzyme (E1) hydrolyzes ATP to form a thioester bond with the C-terminal of ubiquitin. Then, ubiquitin is transferred via the thioester-like complex to the ubiquitin-conjugating enzyme (E2). Finally, the ubiquitin ligase enzyme (E3) mediates the conjugation of the ubiquitin moiety to either a lysine residue or the extreme amino terminus of the targeted protein in a highly controlled manner (Figure 2) [13,14,15,16,17]. All research focused on the ubiquitination cascade and the ubiquitination field was recognized with the Nobel Prize in Chemistry in 2004 awarded to Aaron Ciechanover, Avram Hershko, and Irwin Rose by the Royal Swedish Academy of Sciences.

### 2.1. Number of Ubiquitination Enzymes

Humans have eight E1 enzymes which share the adenylation domain necessary for UBL recognition. These E1 enzymes are found as monomers, heterodimers, and homodimers (Table 1). The human genome encodes two ubiquitin E1s, Uba1 and Uba6. Uba6 conjugates less than 1% of Ub while Uba1 is responsible for the 99% conjugation of cellular Ub [18,19,20]. Uba1 has two isoforms, Uba1A, which is predominantly nuclear, and Uba1B, which is cytosolic [21,22]. The complexity of the conjugation cascade increases for every ubiquitin-like modifier while progressing through the cascade. However, the complexity in ubiquitination can be observed in early stages of the cycle, as the Ub-charged E1 can interact with 38 Ub-E2s differing from other UBLs, where only one E2 has been found [23] (Table 1). Reporting the complexity of ubiquitination, possibilities for substrate conjugation escalate even more when Ub-charged E2 interacts with the E3 enzyme, considering that there are more than 600 different E3 ubiquitin ligases providing substrate specificity [24].

The final conjugation of Ub to the substrate depends on the catalytic structure of the E3 enzyme, which can be classified into three major families: the Really Interesting New Gene (RING) family, the Homologous to E6-associated protein C-terminus (HECT) family, and the RING-in-between-RING (RBR) family (Figure 3).

**RING E3 enzymes**. This is the largest family of E3 enzimes and it is well characterized by its zinc-binding RING domain or by an U-box domain, which adopts the same RING fold but does not contain zinc [32]. This domain facilitates direct Ub transfer from the E2 enzyme to the substrate, functioning as a scaffold to orient the ubiquitin-charged E2 to the substrate protein [33]. RING E3 enzymes can function as monomers (c-CBL), homodimers (RNF4), or heterodimers (BRCA1/BARD1). A noteworthy difference between homodimers and heterodimers is the capacity of homodimers to bind two E2s (one each monomer), while active heterodimers bind only one E2 enzyme [34,35]. RING E3 dimers also differ depending on how they are formed. Dimerization can occur through sequences flanking the RING domain (i.e., BRCA1/BARD1) [36] or sequences within the RING per se (i.e., RNF4) [37]. In the first case, dimers are usually formed via α-helix and in the second case the dimer is formed via interleaved C-termini. In both cases, the two RINGs are positioned allowing the E2-binding surfaces to face away from each other to enable the interaction with the E2 enzyme [38]. Finally, there are some RING E3s that exist as multi-subunit assemblies, such as the Cullin-RING E3 ubiquitin ligases (CRLs). CRLs are assembled on a cullin (CUL) scaffold, containing a globular domain with an embedded RING finger protein (RBX1/RBX2/HRT1) in the C-terminus. This embedded RING finger protein serves as the site for E2 binding and the ubiquitin transfer activity. CRLs also contain an adaptor protein in the CUL N-terminus that binds to distinct sets of substrate receptors (SR), which ensures substrate specificity [39] (Figure 3). A notable example of a large multi-subunit complex E3 is APC/C, which is an assembly of 19 subunits whose cullin-like subunit is Apc2 [40].

**HECT E3 enzymes**. This family is characterized by a conserved HECT domain (350 amino acids), which is located at the C-terminus of the E3 ligase. On the other side, the N-terminus domain is very diverse and mediates substrate targeting. The HECT domain has two well-characterized lobes; the N-terminal lobe interacts with the ubiquitin-charged E2, whereas the C-terminal lobe contains the catalytic cysteine that catalyzes the ubiquitin transfer to the substrate protein in a two-step reaction [41]. First, Ub is transferred from the E2 to the catalytic cysteine of the E3. Secondly, the Ub is conjugated to the target protein from the catalytic cysteine of the E3. In order to facilitate the Ub transfer between the E2 and the catalytic cysteine of the E3, the N- and C-terminal lobes are connected through a flexible hinge that allows them to come together [42]. Based on the N-terminal extensions of these enzymes, HECT E3 enzymes can be classified into three subfamilies: the Nedd4 family, which contains tryptophan–tryptophan (WW) motifs, the HERC family, which possesses one or more (regulators of chromosome condensation 1) RCC1-like domains (RLDs), and the remaining HECT E3 enzymes that contain various domains. The HECT E3s family is often regulated by intramolecular interactions that keep the protein in an autoinhibited state, that is released in response to various signals. An example is Smurf1, a NEDD4 HECT E3 ligase which autoinhibits itself through its C2 domain [43]. 

**RBR E3 enzymes**. Similar to the HECT E3 family, RBR E3s catalyze ubiquitin transfer in a two-step reaction where ubiquitin is first transferred from the E2 to the catalytic cysteine on the E3 and then to the substrate protein [44]. The RBR family is the smallest E3 family and is characterized by the presence of two RING domains (RING1 and RING2) separated by an in-between-RING domain (IBR). RING1 is required for the recruitment of the ubiquitin-charged E2, and RING2 possesses the catalytic cysteine. The IBR adopts the same fold as RING2 but lacks the catalytic cysteine residue. RBR E3s contain additional specific domains that are involved in intramolecular interactions that keep the protein inactive. This inactivation state can be modified by PTMs such as phosphorylation or by protein–protein interactions [45]. Within this family, we can find PARKIN and HOIP as notorious RBR E3 members, respectively, involved in Parkinson disease and being the central catalytic factor of the LUBAC (linear ubiquitin chain assembly complex) [46,47].

The main mechanistical difference is that RING E3s facilitate the direct transfer of ubiquitin from the E2 to the target protein, whilst HECT and RBR E3s contain an active site cysteine that forms a thioester bond with ubiquitin before transferring it to the substrate [7,48,49] (Figure 3). 

### 2.2. Linkages and Cellular Function

After a first ubiquitination cascade, the tethered ubiquitin can become a target for additional ubiquitination, giving rise to ubiquitin polymerization and the formation of polyubiquitination chains and branches. Ubiquitin contains seven acceptor lysines and the amino terminus where chains can be formed (M1, K6, K11, K27, K29, K33, K48, and K63). This offers multiple possibilities to assemble a specific polymer, from monoubiquitination and multi-monoubiquitination to diverse polyubiquitination chains. Polyubiquitination chains can have different types, topologies, and configurations. Ubiquitin chains that comprise only a single linkage type are called homotypic chains (for example a K48 linkage). In contrast, chains that contain mixed linkages are called heterotypic chains. Heterotypic chains can be more complex if one ubiquitin molecule is ubiquitinated at two or more sites creating branches which are known as branched chains (for example K11/K48 linkages) [50]. Depending on the acceptor lysine and the configuration of the linkages, the ubiquitination signal can drive different cellular outcomes (Figure 4). 

The conjugation of a single ubiquitin molecule to one (monoubiquitination) or several lysines (multi-monoubiquitination) is the major ubiquitination event. In yeast, this accounts for over half of all conjugated ubiquitin [51]. Monoubiquitination has a special role in DNA damage repair, signal transduction, and endocytosis [52]. Recently, novel roles in DNA damage repair have been described, being required in DNA crosslink repair by FANCD2 monoubiquitination to promote the closure of FANCD2/FANCI heterodime [53] and DNA trans-lesion synthesis where PCNA monoubiquitination is thighly regulated [54,55]. While the role of monoubiquitination in the DNA damage response has been predominantly studied in histones and associated to malignancies and neurodevelopmental disorders [56,57,58], multi-monoubiquitination has been mainly reported in endocytosis [59]. After the addition of one ubiquitin moiety, monoubiquitination can lead to the formation of ubiquitin chains. 

Early research connected the formation of K48 homotypic ubiquitin linkages to the delivery of a target protein to proteasomal degradation [60]. Later, K11 homotypic chains were discovered to target proteins for proteasomal degradation similarly as K48 chains [61]. In the last decade, K11/K48 heterotypic chains were also reported to not only deliver a substrate protein to proteolytic degradation, but also to be a stronger proteolytic signal [50]. However, non-proteolytic functions have also been elucidated for ubiquitin linkages. For example, polyubiquitination linkages in K63 have been reported to have a key role in several cellular processes including signal transduction, growth response, transcriptional regulation, protein kinase activation, viral protein activation, DNA replication, and DNA repair [62,63,64,65]. K11 linkages have been involved not only in proteasome degradation, but also in several cellular functions with non-proteolytic commitments [66]. 

Together with K48 linkages, K11 and K63 linkages are known as the most abundant ubiquitin chains in cells, being K48 the canonical ubiquitin chains for protein degradation by the proteasome [67,68]. However, the non-conventional or atypical ubiquitin linkages have retained attention in recent years. One of the most studied is the essential role of the PARKIN E3 ligase and its function in regulating mitophagy by atypical K6 linkages [69,70]. In contrast, other E3 enzymes, such as BRCA1/BARD1 and HUWE1 have been linked to K6 linkages, with a role in DNA replication and repair, although the exact function of these K6 linkages remains unclear [71,72,73]. Another atypical linkage involved in DNA repair is K27. RNF168 E3 ligase seems to form K27 polyubiquitin chains to signal DNA damage [74]. The development of chemical probes and affimers against specific linkages in combination with proteomics have brought light to our understanding of cellular function of both K6 and K27 linkages [73,75]. Other atypical ubiquitin linkages occur at K29 and K33. The K29 linkages seem to have a role in embryogenesis and tumorigenesis [76], while K33 linkages have been associated with the regulation of the immune response [77]. Finally, linear ubiquitination chains at M1 are formed by the linear ubiquitin chain assembly complex (LUBAC) and have been reported to be associated to pathologies and the immune response [78,79].

New cellular functions of heterotypic and branched chains are also being uncovered. Besides the abovementioned K11/K48 heterotypic chains, which have a role in cell cycle regulation [50], other branched chains such as K29/K48, K48/K63 and M1/K63 are associated with endothelial reticulum associated degradation (ERAD) and control of membrane fluidity (OLE pathway), NF-κB signaling, apoptosis, and immune response, respectively [80,81,82,83]. Moreover, proteomic analyses have revealed the existence of K6/K48, K11/K33, K27/K29, and K29/K33 branched chains [73,84,85,86].

Recently, a novel study using *Saccharomyces cerevisiae* as a model organism presented a strategy enabling the formation of tailored linkage-specific ubiquitin chains on targeted substrates [87]. Specifically, they studied the consequences of modifying PCNA with different ubiquitin chains. While K63 or M1 ubiquitin chains on PCNA contributed to error-free template switching (TS), K48 polyubiquitination linkages on PCNA induced its degradation by the proteasome. These approaches open new opportunities for the study of specific chain-on-substrate consequences for protein and pathway fate. 

### 2.3. Crosstalk with Other Ubls

This complex ubiquitin signaling system controls virtually all cellular functions and acts as “the ubiquitin code”. It was previously described by Komander and Rape, and it is composed by “writers”, the E1-E2-E3 enzymes, “readers”, the proteins that recognize ubiquitinated proteins by their ubiquitin-binder domains (UBDs), and “erasers”, the deubiquitinating enzymes (DUBs) that can disassemble ubiquitin chains [88].

The complexity of the code increases even more when different PTMs are involved in the same process. Ubiquitin can also be modified by acetylation, phosphorylation, ADP-ribosylation, phosphoribosylation, deamidation, succinylation, and SUMOylation (Kliza 2020). A suitable example of Ub modification is the S65 ubiquitin phosphorylation, which is mediated by PTEN-induce putative kinase 1 (PINK1). This Ub modification mediates substrate specificity and unlocks the autoinhibition of PARKIN E3 ligase [89]. Besides free Ub modification, the crosstalk between ubiquitination and other PTMs during cellular processes boosts the spread, subtlety, and complexity of the ubiquitin code. In fact, neddylation is involved in the ubiquitination process itself. As was shown by cryo-EM, Nedd8 coordinates ubiquitination in Cullin-RING E3 enzymes by binding CUL1 and forming a globular activation module which helps the recruitment of the E2 enzyme for subsequent Ub transfer to the substrate. The presence of Nedd8 stimulates the reaction 2000-fold comparing to the absence of Nedd8, indicating the need of this PTM for the resulting ubiquitination [10]. Similar to Nedd8, other ubiquitin-like proteins like SUMO are able to coordinate ubiquitination responses [9,90].

## 3. SUMOylation

Small Ubiquitin-like Modifiers (SUMOs) share a similar three-dimensional structure with other UBLs. However, SUMOs differ due to their flexible N-terminus, which also contains the site for SUMO chain formation. All eukaryotes express at least one SUMO paralogue. Five SUMO family members have been identified in humans (SUMO1, SUMO2, SUMO3, SUMO4, and SUMO5) [91]. However, SUMO1, SUMO2, and SUMO3 are the main family members where they are commonly classified as SUMO1 and SUMO2/3 because of the high similarity between mature SUMO2 and SUMO3. All SUMO paralogues are similar in structure but differ in expression levels. SUMO2 is the most abundant family member in mammalian cells. Studies in mice show that the knockout of SUMO2 is embryonic lethal, while SUMO1 and SUMO3 knockout mice were associated to mild phenotypes, possibly because SUMO2 might compensate the loss of either SUMO1 or SUMO3 [92,93]. In contrast to ubiquitination, SUMOylation occurs predominantly in the nucleus and is involved in all nuclear processes. Mis-regulation of these UBLs (Ubiquitin and SUMOs) are connected to diseases including cancer [94,95].

The conjugation of any SUMO member to a substrate is termed SUMOylation. Similarly to ubiquitin, SUMO is conjugated in a 3-step enzymatic cascade that involves a dimeric E1 activating enzyme (SAE1 and SAE2), an E2 conjugating enzyme (Ubc9), and several SUMO E3 enzymes (Figure 5). Unlike Ub, each SUMO is encoded by one functional gene, which is translated into a premature SUMO moiety (pre-SUMO) [96]. The pre-SUMO matures with the action of SUMO specific proteases known as SENtrin-specific Proteases (SENPs) [97]. The SENPs remove C-terminal amino acids in order to expose the C-terminal diGly motif, which is needed for the conjugation to the specific lysine of a substrate protein. In vitro experiments have shown that high concentration of Ubc9 and the E1 enzyme is often sufficient to SUMOylate a substrate protein [98,99]. However, in vivo, most substrates require the presence of a SUMO E3 enzyme, which facilitates the transfer of SUMO to the acceptor K of a substrate by enhancing the interaction between the SUMO-charged Ubc9 and the substrate protein [100].

In contrast to the ubiquitin system where more than 600 ubiquitin E3 enzymes have been elucidated, only a few SUMO E3 enzymes have been discovered. SUMO E3 activity in vivo and in vitro was observed for first time in 2001, where SIZ1 and SIZ2 SUMO E3s were required for most SUMO conjugation in *yeast* [101] and PIAS1 acted as a SUMO E3 towards p53 in human cells [102]. In mammals, unrelated classes of proteins appear to act as SUMO E3 enzymes, including the protein inhibitor of activated STAT (PIAS) family, the nuclear pore complex (NPC) component RanBP2, the zinc finger 451 (ZNF451) class, the SLX4 complex and other possible SUMO E3 enzymes that enhance SUMOylation of one or more substrates such as the human polycomb protein Pc2/CBX4, topoisomerase I binding protein (TOPORS), the transcription factor Krox20, the tumor suppressor p14/Arf, the histone deacetylase HDAC4, and the Ras homologue enriched in striatum (Rhes) which must be further evaluated (Table 2).

### 3.1. SUMO E3 enzymes

The major class of SUMO E3 enzymes is the PIAS family composed of five members (Table 2). This family is characterized by its Siz/Pias Really Interesting New Gene (SP-RING) domain that binds to the SUMO E2 enzyme Ubc9. The five members share structural similarity and act in a similar manner as RING E3 ubiquitin ligases [103]. However, in contrast to Ub RING E3 enzymes, knockout studies in mice showed that PIAS SUMO E3 enzymes seem to be redundant and not essential. Mice displayed mild phenotypes, indicating that the lack of one member of the PIAS family is either dispensable or compensated for other members of the PIAS family [121,122]. The substrate specificity that Ub RING E3 enzymes exhibit is yet to be questioned in the PIAS family.

On the other side, the nucleoporin RanBP2/Nup358 does not contain the SP-RING domain, but instead can form a complex with RanGAP1, Ubc9, and SUMO1 to enable E3 ligase activity [123]. This component of the NPC appears to have different roles during the cell cycle. RanBP2 enriches at kinetochores and the mitotic spindle having essential functions in nucleoplasmic transport during interphase [124] and chromosome segregation mitosis [125]. In addition, new functions are emerging in DNA damage, as recent research has reported the E3 activity of RanBP2 in DNA polymerase lambda SUMOylation [110]. All these functions match with knockout mice studies showing that RanBP2^−/−^ mice were unviable, highlighting the essential role of RanBP2 [126].

Another class of SUMO E3 enzymes was discovered recently. The zinc finger ZNF451 family is composed of ZNF451 isoform 1 (ZNF451-1), isoform 2 (ZNF451-2), isoform 3 (ZNF451-3), and the primate-specific KIAA1586. ZNF451-1 and ZNF451-2 are very similar in contrast to the distant members ZNF451 isoform 3 (ZNF451-3) and KIAA1586. All members share a practically identical N-terminus that includes catalytic tandem SUMO interactive motifs (SIMs) which are necessary for SUMO conjugation. Both SIMs work together, the first SIM places the donor SUMO, while a second SIM binds SUMO on the back side of the E2 enzyme for subsequent SUMO conjugation to the substrate protein [112,127]. In contrast to the N-terminus, the C-terminus differs between family members. ZNF451-1 contains C_2_H_2_-Zinc finger domains, whereas ZNF451-2 lacks residues 870-917 due to alternative splicing. ZNF451-3 encodes a C-terminal deletion of 933-1061 and holds a LAP2alpha domain which is not present in the other family members [128]. The ZNF451 class seems to be inefficient in the initial conjugation of SUMO, although this tandem-SIMs region is sufficient to extend a SUMO chain and form a SUMO polymer. This activity is referred to as E4 elongase. In addition, the ZNF451 class has been implicated in SUMO2/3 polymers formation during both proteasome inhibition and DNA damage stresses [112]. Years later, due to its role in DNA-Protein crosslink repair by stalled TOP2 SUMOylation, it was suggested to re-name this ZNF451 class “ZATT” (zinc finger protein associated with TDP2 and TOP2) although these are probably not the only substrates [129].

SLX4 contains a BTB domain and three putative SIMs essential for SUMO binding and SUMOylation. The BTB domain seems to be important for protein–protein interaction and oligomerization necessary for the formation of the SLX4 complex. Pull-down experiments employing SLX4 SIMs mutants show the capacity of SLX4 to SUMOylate xeroderma pigmentosum group-F (XPF). Interestingly, SLX4 can SUMOylate itself with both SUMO1 and SUMO3. However, SLX4 seems to preferentially use SUMO3 for XPF SUMOylation. SLX4 SUMO E3 activity plays a role in response to global and local replication stress [114].

Additional E3 enzymes have been identified in other organisms, such as herpesvirus where they have a possible role during infection. Examples are SM, UL69, and UL54. Interestingly, the SM and UL69 show preference for SUMO1 and UL54 for SUMO2 [130]. There is still a lot to discover and research to be done, but a complex network could be emerging where SUMO E3 enzymes use different SUMO modifiers to form different chains, in order to lead a substrate protein to a particular function in the cell, which seems to be tightly regulated.

### 3.2. SUMO Polymers

The discovery of the ZNF451 class and its E4 elongase activity gave rise to the study of a possible physiological role of poly-SUMOylation, although the knowledge about SUMO chains signaling remains limited compared to the ubiquitin chain field. A notable difference between SUMO1 and SUMO2/3 polymers resides in the ability to form SUMO chains.

In contrast to SUMO1, SUMO2 and SUMO3 possess a K11 in their flexible N-terminus, which is located in a sequence motif, ψKXE, where ψ represents a large hydrophobic amino acid and X any amino acid. This sequence is referred to as SUMO consensus motif. The consensus motif is preferentially targeted for SUMOylation and it is also present in other potential SUMOylation target proteins [131]. This K11 in the SUMO consensus motif allows SUMO2/3 to form K11 SUMO chains [132]. Although this K11 seems to be the main site for SUMO chains, site-specific mass spectrometry approaches have revealed several other SUMO acceptor lysines within SUMO1, SUMO2, and SUMO3 [133]. For example, SUMO1 contains an inverted SUMO consensus site, ExK [134], and harbors an N-terminal K7, which tolerates low efficient SUMO1 chains formation as demonstrated in vitro and in vivo by site-specific mass spectrometry [133,135]. However, it seems that SUMO1 works as a capping factor, terminating SUMO2/3 chain formation [132,136].

The main consequence of SUMO conjugation seems to be the alteration of binding surfaces of the substrate protein, which can either hinder or promote intra- or intermolecular interactions. Additionally, SUMO is able to promote molecular interaction due to its affinity to SIMs. SIMs are short peptide sequences mostly located in unstructured regions of the modified protein or interacting proteins [137]. These SIMs allow non-covalent interaction between SUMOylated proteins and effector proteins which contain SIMs [138]. Given the fact that SUMOylation occurs predominantly in the nucleus and nuclear bodies, its role varies from transcription regulation and chromatin remodeling to DNA repair and cell cycle progression [139]. Although SUMO polymers have been previously reviewed [140,141], to date there is no indication that different SUMO chain linkages fulfil distinct roles within human cells. However, work in *S. pombe* revealed the possible role of two different SUMO chain linkages (K14 and K30) in response to replication arrest [142].

### 3.3. SUMO and Ubiquitin Crosstalk

SUMO chains on target substrates can be a signal for the recruitment of SUMO-targeted ubiquitin ligases (STUbLs), leading to a crosstalk between SUMOylation and Ubiquitination. STUbLs, such as the RING-finger protein 4 (RNF4), contain a RING domain that binds to an E2 ubiquitin enzyme, and SIMs that allow the interaction with SUMOylated substrates and increase the preference for SUMO modified targets. RNF4 possesses at least three closely spaced SIMs and shows a clear preference for substrates that are modified by a SUMO chain or at least three SUMO moieties [143]. RNF4-mediated ubiquitination results in either K48 or K63 ubiquitin linkages, which, respectively, label the substrate protein for proteasomal degradation or for the recruitment of ubiquitin-binding motif containing proteins. This mechanism has been implicated in a variety of cellular processes, including promyelocytic leukemia (PML) nuclear body (NB) integrity, mitosis, and DNA Damage Response (DDR) [143,144,145,146]. RNF4 regulates DNA damage signaling by controlling the lifetime of proteins involved in DNA repair such as the check point mediator MDC1. It also regulates the whole SUMOylation machinery, E1, E2, and E3s, by labeling the members for proteasomal degradation [9,145,147,148]. However, it is yet to be discovered how only some poly-SUMOylated proteins are targeted by STUbLs and how different STUbLs can bind selective SUMOylated targets.

The SUMO and Ubiquitin interplay can also be modified by SUMO-targeted ubiquitin proteases (STUPs). STUPs can recognize poly-SUMOylated proteins and are able to modify the ubiquitin chains on SUMOylated targets by their ability to remove Ub. To date, three STUPs have been identified. USP7 seems to remove ubiquitin from SUMO targets with a role in DNA replication [149]. USP11 might have a role regulating nuclear bodies by limiting RNF4 activity [150]. The last STUP is Ataxin-3 (ATX3), which seems to participate in the regulation of MDC1 counteracting the RNF4 activity [151]. Finally, there is not only crosstalk with ubiquitination but also with other PTMs. This was observed in a very deep profiling of the SUMO proteome [152].

## 4. Proteomics for Substrate Identification

The major challenge to fully understand the role of a specific E3 ligase is to determine its target proteins. Due to the hierarchical structure of the ubiquitination cascade, one E1 enzyme (Uba1/Uba6) can bind dozens of E2, these E2s can bind hundreds of E3 which are responsible for determining the substrate specificity for ubiquitination. Thus, mapping specific targets for a particular E3 ligase became challenging (Figure 6a). In addition, the transient and weak interaction between the E3 ligase and the target, the dynamic and reversible nature of these modifications, the relative low abundance and expression of substrates, the fact that several substrates are labeled for proteasome degradation and rapidly degraded, increase the difficulty of the identification. Moreover, the ubiquitination is often dependent on physiological conditions and spatiotemporal organization. Therefore, many substrate proteins can only be identified upon different cell treatments. Additionally, under a stabilized physiological condition, several E3 enzymes can target individual substrate proteins at different residues, making the identification even more complex.

The vast majority of the E3 enzymes identified in the human genome remain relatively uncharacterized. Generally, the identification and validation of substrates rely on relatively slow, low-throughput biochemical methods reviewed in [153]. Currently, the increasing advances in proteomics and mass spectrometry (MS) provide new approaches to not only identify the ubiquitin and SUMO modified proteins, but also decipher ubiquitination and SUMOylation sites in target proteins [139,154].

### 4.1. Mapping Sites

In order to map ubiquitination sites, many strategies reside in the diGly residue that is covalently attached to the modified lysine of a substrate protein. The serine protease trypsin is commonly used to generate peptides from a protein sample for consequent MS identification. Trypsin cleaves Ub after R74, leaving this remnant diGly at the ubiquitinated residue of a substrate protein. Immunopurification methods using monoclonal antibodies against selective diGly residues have been exploited for the MS identification of ubiquitination sites [155,156]. However, in addition to ubiquitin, other UBL-modifiers (ISG15 and NEDD8) also leave the diGly residue after trypsin digestion, which is one of the major limitations of these approaches.

In 2018, Blagoy Blagoev’s group developed the UbiSite antibody which recognizes the 13 amino acids of Ub that remain covalently attached to the modified residue of the substrate proteins after digestion with endo-protease LysC. Another advantage of this antibody is that it allows detection of N-terminal ubiquitination but not linear Ub signatures. This method enabled the identification of 63.000 unique ubiquitination sites [154]. More strategies have been recently developed regarding the diGly dipeptide. David Komander’s group established the Ub-clipping approach which utilizes an engineered viral protease (Lb^pro^) to remove the Ub from substrate proteins but leaving the C-terminal diGly conjugated to the modification site. Unlike trypsin digestion, the formed ubiquitin branches can be detected by MS analysis [157].

Immunopurification using antibodies also enables the determination of ubiquitination chains topology. There are Ub linkage-specific antibodies which specifically recognize M1, K11, K27, K48, and K63 linkages [158,159]. More recently, chemical synthesis has enabled the engineering of Ub-binding modules or probes. These probes can be designed as traps for binding specific chains, branches, and even hybrid chains that allow their enrichment for subsequent MS analysis [160,161].

In a similar strategy to the UbiSite, 14869 SUMO2ylation sites were identified at the endogenous level under heat stress and proteasomal inhibition in human cells using the epitope for the commercially available SUMO2 antibody (8A2) [139].

### 4.2. Identification of Novel Substrates

Likewise, MS-based approaches have been performed to identify novel ubiquitination substrates. We could classify these into undirect and direct MS-based approaches (Figure 6b). Within the undirect group, finding differences in ubiquitinated proteins upon either overexpression or depletion of an E3 ligase of interest has been commonly used [162,163,164,165]. Substrate proteins that are enriched or depleted in their ubiquitination levels are considered as putative ubiquitination substrates for the E3 ligase under investigation. However, these indirect approaches are based on full ubiquitin proteomes once the E3 ligase of interest is either overexpressed or depleted. Due to the complexity of the ubiquitin proteome and the low abundance ubiquitination targets, some results might be obtained because of overexpressed artifacts in the case of overexpression of an E3 ligase. Conversely, in knockdown-based screenings, E3s can be redundant on their targets, and some targets might be missed because their ubiquitination is still performed by another E3 enzyme resulting from an epistatic effect. As a consequence, every potential target must be carefully validated.

Regarding direct approaches, several methods have been employed including Ub ligase substrate trapping [166,167], Michaelis intermediates traps [168], NEDDylator approaches [169], Ub activated interaction traps, UBAITs [170], and optimized methods such as Targets for Ubiquitin Ligases Identified by Proteomics, TULIP and TULIP2 [9,148] which allow the identification of E3 ligase-specific ubiquitination targets by mass spectrometry analysis.

**Ligase trapping** is based on E3 enzymes fused to Ubiquitin Binding Domains (UBDs) that are used for isolation of ubiquitinated substrates by affinity purification. The rationale resides in the fact that UBDs increase the binding affinity of the E3 ligase of interest toward its targets and enabled the identification of potential substrates by MS analysis. The selection of proper UBD, as well as the fusion point, are the major limitations of this approach, as effective enrichment of substrates and potential disruption of the substrate recruitment are essential for the proper functionality of the ligase trap [166,167].

**Michaelis intermediate strategies** are based on the generation of a E2-SUMO thioester mimetic than can be crosslinked to the substrate protein. The resulting E2-SUMO-Substrate complex can be purified and the structure can be determined by crystallization [168].

**The NEDDylator approach** relies on the fusion of the NEDD8 E2 enzyme (Ubc12) to the substrate-binding region of the E3 ligase of interest. Such configuration allows artificial NEDDylation of endogenous E3 ligase substrates. The enrichment and subsequent MS analysis relies on the NEDDylation proteome which is far less complex than the ubiquitination. As NEDDylation does not occur at a high level in a cell, it is possible to distinguish between endogenous and NEDDylator-induced modifications to identify E3 ligase substrates [169].

**UBAIT methodology** allows identification of substrates for both RING and HECT E3 enzymes. The UBAIT tool consists of the utilization of E3 enzyme–ubiquitin fusions. The fusion of Ub to the E3 ligase abolishes the transient interaction between the E3 ligase and the substrate, since the E3 ligase remains fused to the Ub which is covalently bound to the substrate protein after its putative ubiquitination. The presence of Ub enables E1- and E2-mediated activation of UBAIT and subsequent covalent capture of E3 ligase substrates. This allows the later purification of the E3 together with its ubiquitination target for subsequent identification by MS analysis. The UBAIT strategy has been later optimized to enable the systematic identification of Ubiquitin E3 substrates, in the TULIP and TULIP2 methodologies [9,148].

### 4.3. Other Ub/SUMO-Related MS-Based Proteomics Approaches

We can also find methods using **proximity****-dependent biotin identification (BioID)**, which can be used to identify weak/transient protein interactions of proximal/surrounding proteins of a particular E3 ligase. It is based on the fusion of the E3 ligase with the mutated form of biotin ligase (BirA), which biotinylates all proteins in the close vicinity, if biotin is available. These biotinylated proteins can be enriched by affinity purification and identified by MS analysis, where the labeled proteins can be potential E3 substrate targets [171]. Recently, this approach has been updated for the identification of SUMO-dependent interactors [172]. In this strategy, TurboID (upgraded version of BioID) was split in two, one fragment was fused to SUMO and the complementary fragment to a protein of interest. When the protein of interest and SUMO interact, the TurboID enzyme can be reconstituted and able to label proximal complexes, that can be later purified and identified by MS.

The identification of a substrate can be even more complex when two different E3 enzymes work together for the ubiquitination of a particular substrate. It is known that many neddylated culling-RING E3 enzymes (CRLs) and RBR enzymes in the ARIH family form E3-E3 super-assemblies [173], hence, new activity-based chemical probes that enable cryo-electron microscopy visualization of E3-E3 ubiquitination have been developed, facilitating the visualization of the ubiquitination intermediates [174].

Some of these strategies can be modified for SUMO targets. In a recent study, PIAS1 was overexpressed to produce profile changes in protein SUMOylation. This profiles can be determined by MS [104]. This is an example of an indirect approach for SUMOylation targets.

It has been shown that defects in ubiquitin and SUMO E3 enzymes may be contributing factors in cancer [175,176,177] and human neurodegenerative diseases such as Alzheimer’s, Huntington’s, amyotrophic lateral sclerosis, and Parkinson’s [178]. Therefore, it becomes crucial to identify substrates that are ubiquitylated/SUMOylated by specific E3 enzymes, in order to obtain novel insights in particular diseases.

## 5. Conclusions and Future Perspectives

During recent years, new tools and techniques have allowed us to exploit the potential of already known features or to discover new ones. A good example of this was the observation of degradation signals in short-lived proteins in 1986 [179]. Nowadays, the use of MS approaches makes it possible to characterize a phosphodegron site, the E3 ligase responsible for the ubiquitination and, with biochemical validation, to know the biological processes this degradation signal is involved in [180].

New tools to explore ubiquitin and SUMO are ubiquitination and SUMOylation inhibitors. Recently, a new SUMO inhibitor has been discovered [181] with optimistic results in the treatment of cancer [182,183]. The employment of inhibitors combined with MS have revealed a large set of SUMOylation and ubiquitination targets. The inhibition of either ubiquitination or SUMOylation in cells stably expressing tag-ubiquitin or tag-SUMO, facilitates the identification of these targets when comparing the treated cells with non-treated ones [184,185].

Another key strategy for the decade ahead is the use of CRISPR genome-wide screening combined with either ubiquitination or SUMOylation inhibitors to study up- and downregulated genes upon a desired condition [186,187].

To get deeper insight into the ubiquitin code and decipher the complexity of the ubiquitination network, the use of chemical synthetic probes combined with high resolution MS equipment would be a key topic for the next decade. Using these probes, it would be possible to find particular ubiquitin or SUMO linkages/branches, or even hybrid chains between SUMO and ubiquitin [188,189,190]. Some strategies have been already reviewed in the ubiquitin filed [191], nonetheless, they can be extended for the SUMO field.

## Figures and Tables

**Figure 1 ijms-23-03281-f001:**
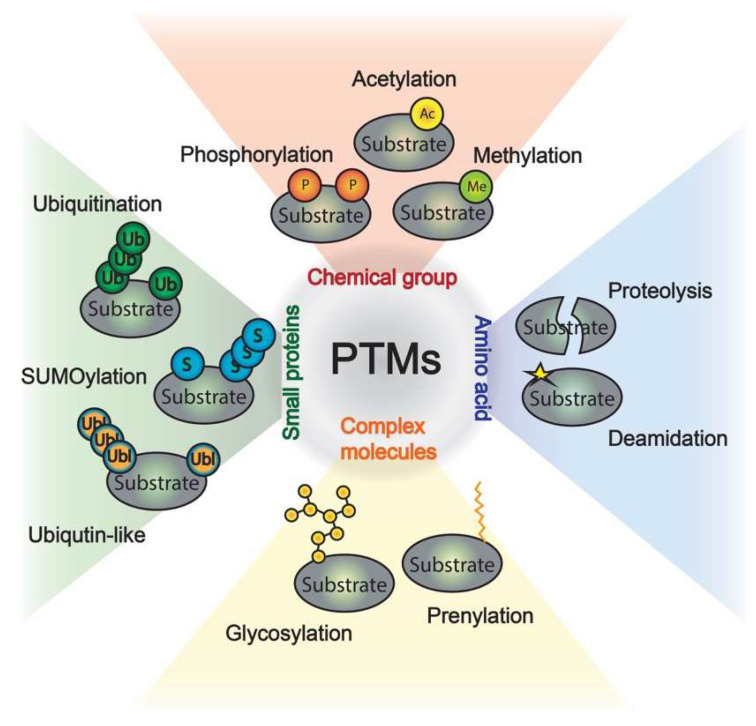
Post-translational modifications of proteins. Chemical group modifications are shown in red, amino acid modifications in blue, complex molecules are in yellow, and additions of small proteins are displayed in green.

**Figure 2 ijms-23-03281-f002:**
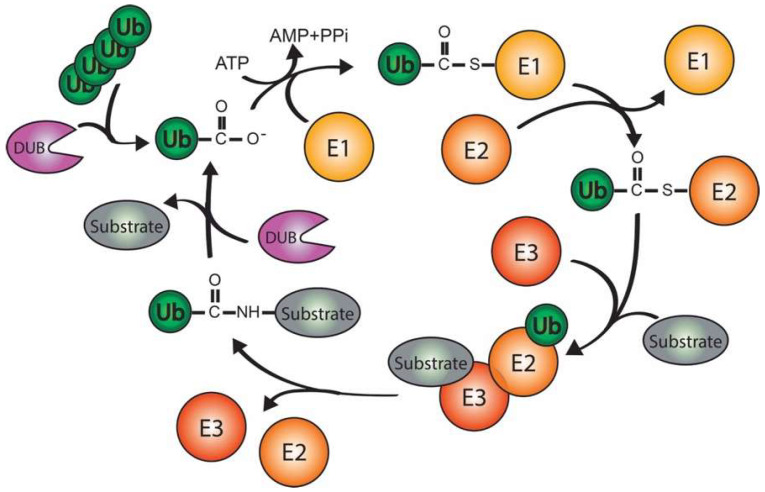
Ubiquitination cascade. Free and active ubiquitin (Ub) is conjugated to the activating enzyme (E1) in an ATP-dependent manner. Then, Ub is transferred to the conjugating enzyme (E2) to be finally covalently attached to the substrate protein assisted by the ligating enzyme (E3) which provides substrate specificity. Subsequently, Ub can be deconjugated from substrates by DeUBiqutinating enzymes (DUBs).

**Figure 3 ijms-23-03281-f003:**
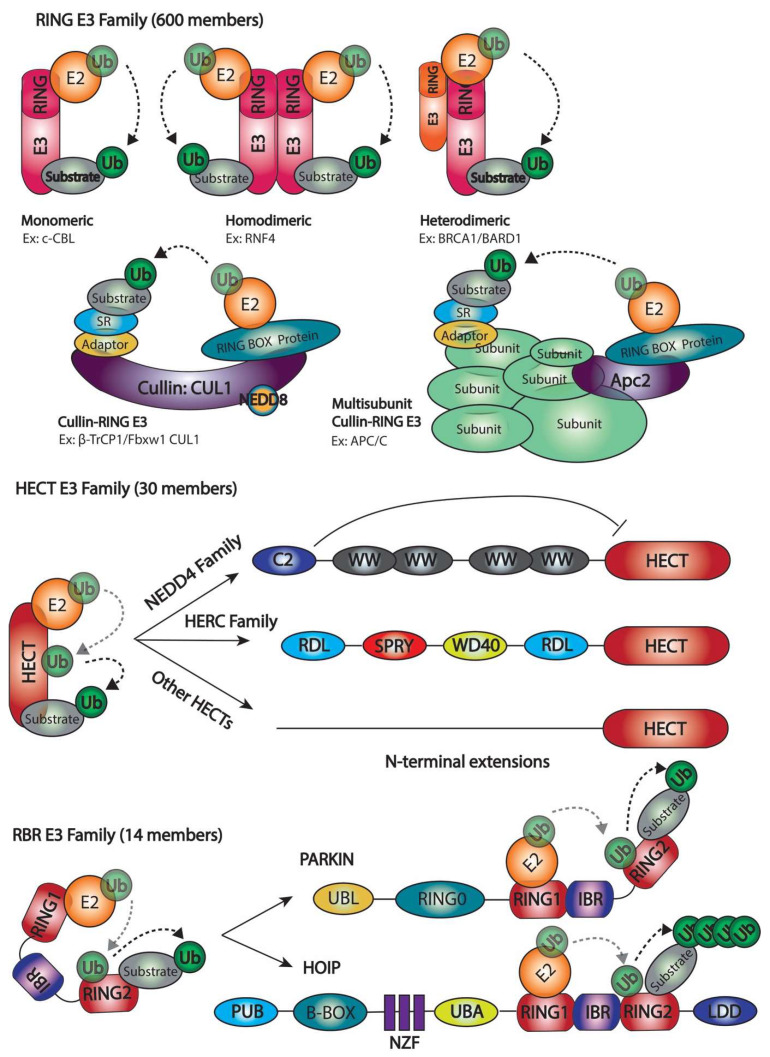
The E3 enzyme families and subclasses. RING E3 enzymes are shown as monomeric, homodimeric, heterodimeric, and forming multi-subunit complexes (CULLINs) E3 enzymes. HECT E3 enzymes are exhibited in three sub-families: NEDD4 Family, HERC Family, and Other HECT. RBR E3 Family is a 14 members family where two members are shown (PARKIN and HOIP), and the ubiquitination mechanism is displayed. The ubiquitination process is depicted. While the Ub transfer from the E2 to the substrate occur in a 2-step reaction for HECT and RBR families, there is a direct ubiquitin transfer from the E2 to the substrate in the RING E3 family.

**Figure 4 ijms-23-03281-f004:**
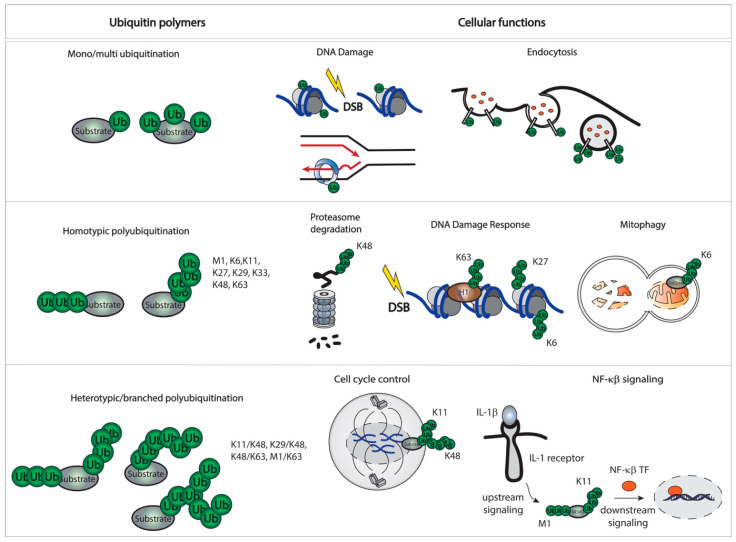
Ubiquitination polymers. Ub moieties can modify proteins at one (mono ubiquitination) or several (multiple mono ubiquitination) Lys residues. Ub can form eight distinctive homotypic linkages, either through M1 (linear Ub chain) or 7 internal Lys residues (K6, K11, K27, K29, K33, K48, and K63 Ub chains). Additional complexity is achieved through the formation of heterotypic Ub chains, which contain multiple Ub linkages and adopt mixed or branched topology. Cellular functions associated to these ubiquitin polymers are displayed.

**Figure 5 ijms-23-03281-f005:**
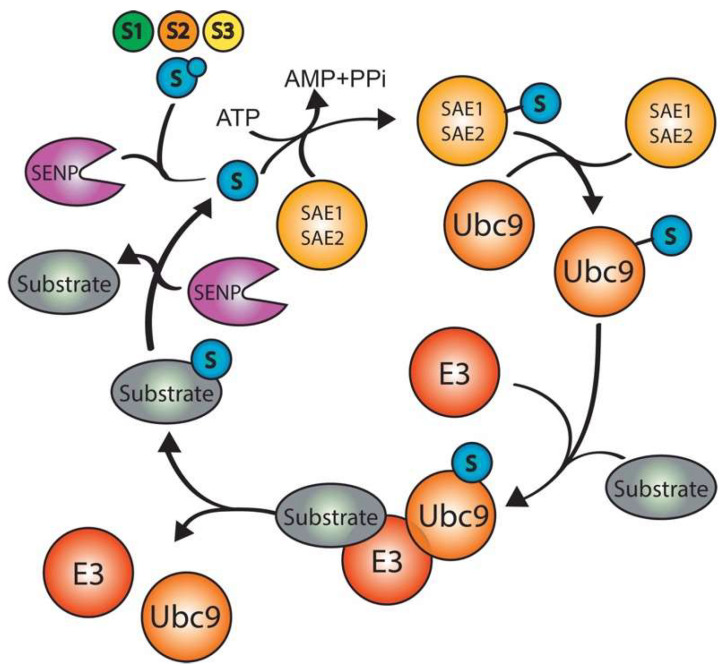
SUMOylation cascade. SUMO precursor matures by the action of a SENP that cleavages the SUMO C-terminal, leaving a diGly motive that forms a thioester bond with the activating enzyme E1 in an ATP dependent manner. Then, activated SUMO is transferred to the conjugating enzyme E2. Finally, the E2 conjugates SUMO to the acceptor lysine (usually in the consensus motive ψKxE) with or without the ligation enzyme E3 which confers substrate specificity. Additional rounds of this cascade form SUMO polymers that can be cleaved by specific SENPs.

**Figure 6 ijms-23-03281-f006:**
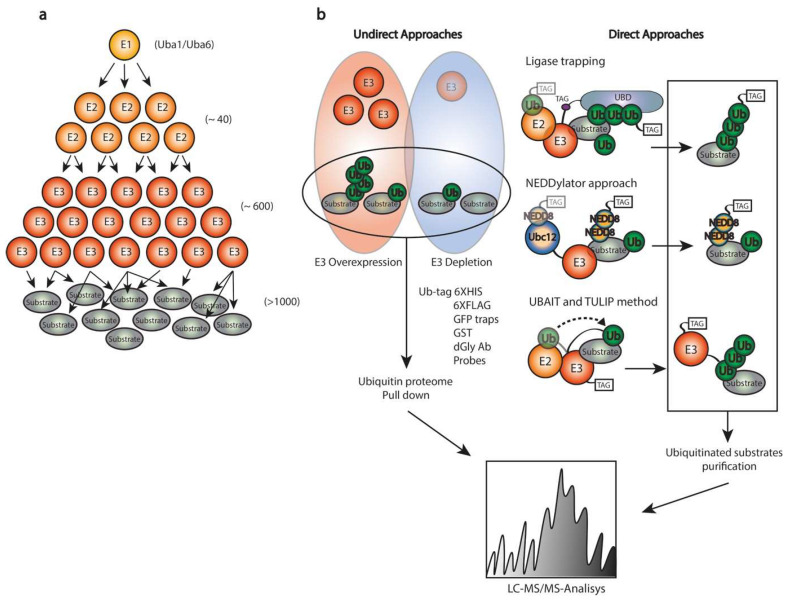
Proteomics for E3 ligase target identification. (**a**) Hierarchical organization of the ubiquitination cascade. Emphasizing the difficulty of mapping substrate proteins for specific E3 enzymes. (**b**) Strategies for E3 ligase substrate identification divided into undirect and direct approaches. Within undirect methods, in red, the overexpression of an E3 ligase results in the increase in ubiquitination levels for putative substrates. Opposing, in blue, the depletion of an E3 ligase displays a decrease in ubiquitination levels of putative substrates. The direct approaches allow identification of specific-E3 ligase substrates where ligase trapping, NEDDylation approach, and UBAIT/TULIP methodology are shown.

**Table 1 ijms-23-03281-t001:** Summary of E1 and E2 enzymes for ubiquitin-like modifiers.

E1	Complex Formation	UBL	Cellular Process	E2	Reference
**Uba1** **Uba6**	Monomer Monomer	Ub	Ubiquitination	38	[23]
**Uba7**	Monomer	ISG15	ISGylation	Ubch8	[25]
**Sae1/Uba2**	Heterodimer	SUMO	SUMOylation	Ubc9	[26]
**Nae1/Uba3**	Heterodimer	NEDD8	NEDDylation	Uba12	[27]
**Uba4/Mocs3**	Homodimer	URM1	URMylation	Unknown	[28]
**Uba5**	Homodimer	UFM1	UFMylation	Ufc1	[29]
**Atg7**	Homodimer	ATG8/ATG12	Autophagy	ATG3/ATG10	[30,31]

**Table 2 ijms-23-03281-t002:** SUMO E3 enzymes.

Class	E3 Ligase	SUMO	Substrates	Chains	Functions	Reference
**PIAS**	PIAS1	SUMO 1, SUMO2/3	p53, PCNA, Vimentin, etc.	K?	Check points regulation, DNA damage, cell migration	[102,103,104]
	PIAS2	SUMO1	p53	K7?	Check points	[105]
	PIAS3	SUMO2/3	ATR	K11?	DNA damage	[106]
	PIAS4	SUMO1, SUMO2/3	LEF1	K11?	Wnt-signaling, DNA damage	[107]
	NSMCE2/NSE2	SUMO1	Rad18, TRAX, MMS21…	K7?	DNA damage	[108,109]
**NCP**	RanBP2/Nup358	SUMO 1	Sp100, Top2, Borealin, DNApolα	K7?	Nuclear import, mitosis, DNA repair	[99,110,111]
**ZNF451**	ZNF451-1/2	SUMO2/3	MCM4, PML	K11	PML stability	[112,113]
	ZNF451-3	SUMO2/3	MCM4	K11		[112]
	KIAA 1586	SUMO1, SUMO2/3	MCM4	K7? K11		[112]
**SLX4**	SLX4 Complex	SUMO1, SUMO2/3	SLX4, XPF	K?	Genome maintenance, cell division	[114]
**Other SUMO E3 enzymes**	Pc2/CBX4	SUMO1	CtBP	-	Polycomb bodies	[115]
	TOPORS	SUMO1	p53	-	Check points regulation	[116]
	Krox20	SUMO1	Nab	-	Krox20 autoregulation	[117]
	p14/Arf	SUMO1, SUMO2/3	MDM2, NPM/B23	K?	Tumor suppression	[118]
	HDAC4	SUMO1	MEF2	Mono	Muscle cell differentiation	[119]
	Rhes	SUMO1, SUMO2/3	Ubc9	Multimono?	SUMOylation	[120]

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
