# Peer review of "Insights in Post-Translational Modifications: Ubiquitin and SUMO"

_ijms, 2022, doi:10.3390/ijms23063281_

Round 1
Reviewer 1 Report
Remarks to the Author:
In this review, the authors extensively discussed the basic mechanisms of both ubiquitination and SUMOylation. In addition, they all described how recent mass-spectrometry (MS)-based approaches contribute to the understanding of ubiquitination and SUMOylation, especially for finding special targets of particular E3 enzymes.
Overall, it’s nice review. The authors have very strong background in the flied of PTM. Very unfortunately, the figures are missing. I strongly suggest the authors double-check the manuscript before the last submission.
Major comments:
1. I try to find the figures. Unfortunately, I could not find them.
2. I suggest the authors combine figure 2 and figure 3 if it’s possible.
3. line 167, “figure 4” should be “Figure 4”
4. Title for table 2 is misplaced.
Author Response
We are happy to read that this reviewes supports publication of our manuscript and appreciatte his/her comments.
We have addressed the typos mentioned buy the reviewer and included now the figures in the .docx file. Figures were previously in a separate PDF file included in the .zip file together with the manuscript.
Although we appreciate the suggestion to merge figures 2 and 3, we consider that they will look better as separate. Hopefully this reviewer will agree with us once the figures are visible.
Reviewer 2 Report
This review article titled “Insights in Post-translational Modifications: Ubiquitin and SUMO” gives a concise overview of the key enzymes (E3s) involved in the ubiquitination and sumoylation of protein substrates. Moreover, the review also summarizes recent MS-based approaches to define targets for E3-ligases.
I would suggest including a table format to list all linkages/topologies with references discussed in the review, just like table for E3s. Somehow, I did not find figures included in the downloaded copy.
In my opinion the subsections are appropriately chosen and well written. The review should serve as a valuable resource for the scientific community interested in this topic, and other non-field readers in general.
Author Response
We thanks this reviewer for the support for publication of our review.
We consider that Figure 4 would suffice and is also more illutrative than a table full of numbers of references.
We regret that this reviewer could not visualize the figures which were included as a separate PDF file. We have now included the figures in the .docx file and moved the figure legends accordingly.
Round 2
Reviewer 1 Report
The authors has addressed all my concerns. I recommend acceptance in the current form.